# ZERO-SHOT HUMAN-OBJECT INTERACTION DETECTION VIA CONDITIONAL MULTI-MODAL PROMPTS

## ABSTRACT

Human Object Interaction (HOI) detection is the task of locating and inferring the relationships between all possible human-object combinations. One of the most challenging issues is the extensive labor required for the annotation of combinatorial space of possible HOI interactions. Most existing HOI detectors rely on full annotations of all predefined interactions, resulting in a lack of generalisation for unseen combinations and actions. Inspired by the powerful generalisation ability of the large Vision-Language Models (VLM), we propose a Prompt-based zero-shot human-object Interaction Detection framework, namely PID, which can improve alignment between the vision and language representations using conditional multi-modal prompts. Specifically, different from traditional prompt-learning methods, we propose learning decoupled visual and language prompts for spatial-aware visual feature extraction and interaction classification, respectively. Furthermore, we introduce constraints for multi-modal prompts to alleviate the problem of overfitting to seen concepts in prompt learning process, thus improving the suitability for zero-shot settings. Extensive experiments demonstrate the prominence of our detector with conditional multi-modal prompts, outperforming previous state-of-the-art on unseen classes of various zero-shot settings.

## 1 INTRODUCTION

Human-object interaction (HOI) detection has been introduced by Gupta & Malik (2015) and plays an important role in understanding high-level human-centric scenes. Given an image, HOI detection aims to localize human and object pairs and recognize their interactions, *i.e.* a set of <human, object, action> triplets. Traditionally, human-object interaction detectors can be categorized as one- or two-stage. Two-stage methods (Zhou & Chi, 2019; Li et al., 2019; Liu et al., 2020a;b; Li et al., 2020b; Zhang et al., 2021b; 2022a;b; Liu et al., 2022; Wu et al., 2022b) localize the humans and objects individually using off-the-shelf detectors (e.g., DETR (Carion et al., 2020)), then the region features from the localized area are used to predict interaction class. One-stage methods leverage multi-stream networks (Liao et al., 2020; Wang et al., 2020) or encoder-decoder architectures (Zou et al., 2021; Chen et al., 2021; Kim et al., 2021; Tamura et al., 2021; Zhong et al., 2022; Liao et al., 2022; Ning et al., 2023) to predict HOI triplets from a global image context in an end-to-end manner.

Despite recent advances, most previous works lack generalisability to unseen HOIs. Although some zero-shot HOI detectors (Hou et al., 2020; 2021b; Liao et al., 2022; Wu et al., 2022a; Wang et al., 2022b) have been proposed in recent years, some of them (Hou et al., 2020; 2021b) fail to incorporate language priors and can't generalise to unseen verbs. Besides, as shown in Figure 1a, many previous zero-shot detectors perform significantly worse on unseen classes than on seen classes, which we call "performance degradation". For example, as shown in Figure 1a, the mAP of GEN-VLKT (Liao et al., 2022), EoID (Wu et al., 2022a) and HOICLIP (Ning et al., 2023) on unseen classes is lower than that on seen classes by 9.27%, 8.02% and 7.89%, respectively. Given the combinatorial nature of HOIs, constructing a HOI dataset with all possible HOIs is prohibitively expensive. This motivates us to investigate a HOI detector that can be applied to a wide range of previously unseen interactions with powerful generalisability. Zero-shot HOI detection has the following two challenges: 1) how to extract interactiveness-aware features for human-object pairs in order to determine whether they interact with each other when confronted with unseen HOI concepts, and 2) how to recognize the unseen interaction types accurately.

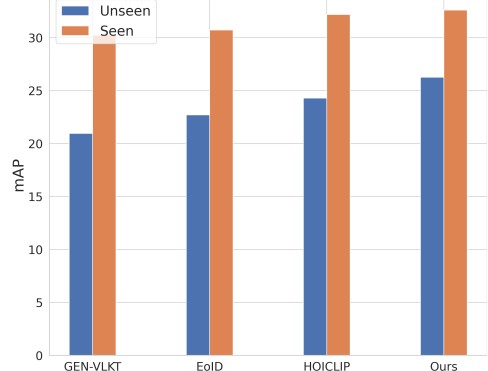
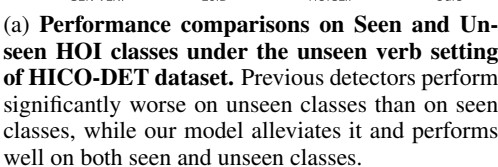
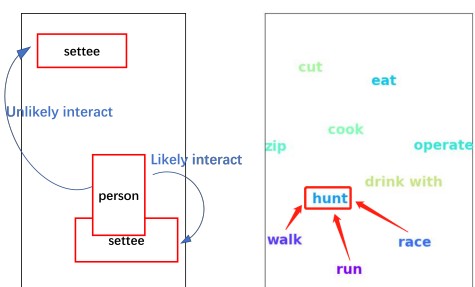

(a) **Performance comparisons on Seen and Unseen HOI classes under the unseen verb setting of HICO-DET dataset.** Previous detectors perform significantly worse on unseen classes than on seen classes, while our model alleviates it and performs well on both seen and unseen classes.

(b) **Left: Unseen concept sit-on-settee.** Spatial cues help recognize interactiveness of unseen HOI concepts; **Right: t-SNE visualization of CLIP embeddings of verbs.** Text helps recognizing interaction categories of unseen verbs, e.g., hunt.

Figure 1: **Zero-shot HOI detection on the unseen verb setting.** Conventional detectors' performance has a clear gap between seen classes and unseen classes. For example, the mAP of HOICLIP on unseen classes is 7.89% lower, than on seen classes. In contrast, our model uses visual spatial cues during feature extraction to help judge interactiveness and proposes sharing knowledge across verbs.

To address the aforementioned issues, *we propose PID, which divides HOI detection into two subtasks: interactiveness-sensitive spatial-aware visual feature extraction and generalisable interaction classification. The design aids in reducing their dependence on one another and error propagation between them.* We propose decoupled vision and language prompts for the above two subtasks to prevent mutual inhibition, respectively. For the first subtask, we design vision prompts to guide the visual features to be verb-agnostic, so that it can generalise its ability to extract interactiveness-aware features to previously unseen classes as shown on the left of Figure 1b. Specifically, inspired by the generalisation capability on classification tasks of VLMs, e.g., Radford et al. (2021), we employ the attention mechanism (Vaswani et al., 2017) to integrate the knowledge in vision prompts into the image encoder from the early spatial-aware and fine-grained feature maps, where valuable information is lied for HOI detection task. For the subtask of interaction classification, we propose language prompts that are unaware of spatial information. The language prompts provide a unified context for both seen and unseen HOIs, allowing the model to leverage knowledge learnt from seen classes to classify HOIs that include unseen verbs as shown on the right of Figure 1b. The multi-modal prompts serve as a hub to build a connection between seen and unseen categories.

Furthermore, the proposed multi-modal prompts need to condition on human-designed prior knowledge to alleviate the problem of overfitting to the seen categories. To further avoid the mutual dependence, we propose an approach where the prompts for the two modalities are independent of each other as illustrated in Figure 2. Each prompt is tailored to leverage distinct types of prior knowledge, ensuring a more diversified and robust learning process. For the vision prompts, we propose input-conditioned instance-level prior knowledge to help treating the potentially interactive instances belonging to seen and unseen categories equally. The instance-level prior knowledge contains spatial-aware and interaction-invarient information of the given image. For the language prompts, to make use of the prior of the text space of the large Vision-Language Models, we use human-designed prompts as a regularizer to keep the learned text prompts from diverging too much.This constraint preserves the origin semantic space learnt by VLM, and thus may be better for potential real-world scenario applications where arbitrary novel actions may occur.

We evaluate our detector with conditional multi-modal prompts under various zero-shot settings. Experiments show that our model not only performs well on both seen and unseen classes, but also narrows the performance gap between seen and unseen classes, as shown in Figure 1a.

Our contributions are threefold: (1) To the best of our knowledge, we first propose multi-modal prompts in zero-shot human-object interaction detection to improve visual-language feature alignment and zero-shot knowledge transfer. (2) In order to alleviate the problem of overfitting to seen concepts and further improve the model's generalisation ability, we further propose separate conditions for both modalities. (3) Our model sets a new state-of-the-art for HOI detection on unseen classes in various zero-shot settings, significantly outperforming all previous methods.

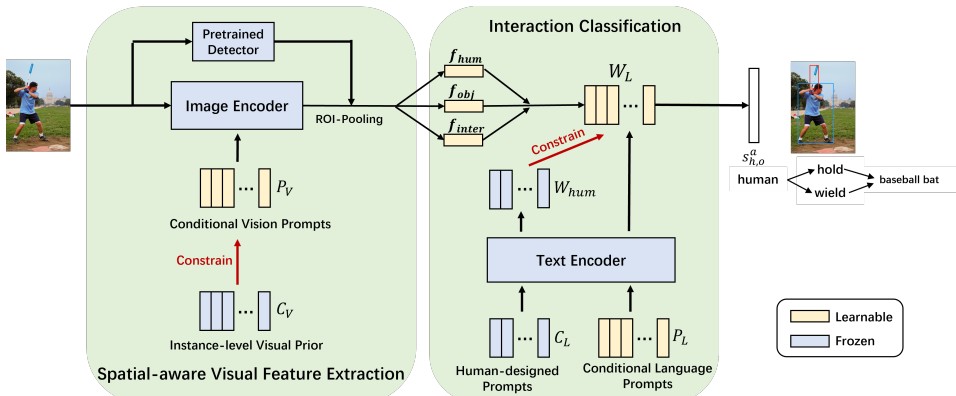

Figure 2: **The overall framework of PID.** The proposed method splits zero-shot HOI detection into two subtasks: spatial-aware visual feature extraction and interaction classification. We propose decoupled visual and text prompts for each subtask to eliminate the dependence between them and breaks error-propagation in-between. The conditional vision prompts ($P_V$) are used to inject spatial-aware and verb-agnostic knowledge into the image encoder and are explicitly constrained by instance-level visual prior ($C_V$). The conditional language prompts ($P_L$) are constraint by the human-designed prompts ($C_L$) through a regularization loss. (Best viewed in color.)

## 2 RELATED WORK

### 2.1 HUMAN-OBJECT INTERACTION DETECTION

With the development of large-scale datasets (Gupta & Malik, 2015; Chao et al., 2018; Kuznetsova et al., 2020; Liao et al., 2020) and deep learning-based methods (Li et al., 2020a; Ulutan et al., 2020; Chen et al., 2021), HOI learning has been rapidly progressing in two main streams: one- and two-stage approaches. One-stage HOI detectors usually formulate HOI detection task as a set prediction problem originating from DETR (Carion et al., 2020) and perform object detection and interaction prediction in a parallel (Chen et al., 2021; Kim et al., 2021; Tamura et al., 2021) or sequentially (Liao et al., 2022). In contrast, two-stage methods (Liu et al., 2020a; Li et al., 2020b; Zhang et al., 2021b; 2022a;b) usually utilize pre-trained detectors (Ren et al., 2015; He et al., 2017; Carion et al., 2020) to detect human and object proposals and exhaustively enumerate all possible human-object pairs in the first stage. Then they design an independent module to predict the multi-label interactions of each human-object pair in the second stage. Despite their improved performance, most of previous models rely heavily on full annotations with predefined HOI categories and thus are costly to scale further. Moreover, they lack the generalisation capatility to deal with unseen HOI categories. In contrast to them, our work target on zero-shot HOI detection with the help of an off-the-shelf object detector and Vision-Language Model in a two-stage manner.

### 2.2 ZERO-SHOT HOI DETECTION

Zero-shot HOI Detection aims at detecting interactions unseen in the training set, which is essential for developing practical HOI detection systems that can function effectively in real-world scenarios. Liu et al. (2020b) converts HOI categories and their components into a graph and distributes knowledge among its nodes. Hou et al. (2020) recombines object representations and human representations to compose unseen HOI samples. Hou et al. (2021b) proposes to generate fake object representations for human-object recombination. Hou et al. (2021a) exploits additional object datasets for HOI detection to discover novel HOI categories. However, lacking of the help of semantics, the above methods aren't capable of detecting HOIs including unseen verbs.

To incorporate language priors in zero-shot HOI detection, Liao et al. (2022); Wu et al. (2022a); Ning et al. (2023) propose to distill knowledge from CLIP (Radford et al., 2021) to achieve zero-shot HOI detection. The nature generalisability of language aids models in recognizing HOIs, even those with unseen verbs. Despite the progressing generalisability, previous methods still lack of proper regularization and thus tend to overfit to seen categories. For example, their performance on unseen classes may be around 10% lower than that on seen classes under the evaluation metric of mAP. Different from Liao et al. (2022); Wu et al. (2022a), our model solves zero-shot HOI detection in a two-stage manner and achieves knowledge sharing between seen and unseen HOIs via conditional multi-modal prompts, resulting in better generalisability to unseen HOI concepts.

## 2.3 Prompt Learning

Recently, the development of large vision-language model (VLM), e.g., CLIP (Radford et al., 2021), emerges and finds its applications in few-shot or zero-shot learning tasks (Zhang et al., 2021c; Gao et al., 2021). Inspired by prompt learning in language tasks, CoOp (Zhou et al., 2022b) first proposes to use context tokens as language prompts in the image classification task. Co-CoOp (Zhou et al., 2022a) proposes to explicitly condition language prompts on image instances. Recently, other approaches for adapting V-L models through prompting have been proposed. MaPLe (khattak et al., 2023) proposes a coupling function to explicitly condition vision prompts on their language counterparts, to provide more flexibility to align the vision-language representations. However, existing methods primarily focus on prompt learning for image classification, which may not be suitable for HOI detection. Liao et al. (2022) and Wang et al. (2022b) first propose applying static template prompts or learnable language prompts in the HOI detection task, respectively. However, they ignore the fact that HOI detection involves considering regional spatial information, making it distinct from image classification. Therefore, how to design tailored *spatial-aware* prompts specifically designed for the HOI detection task is critical. Note that the human-object localisation and interaction classification are distinct subtasks, we propose to employ decoupled multi-modal prompts for the two subtasks to reduce error propagation between them. Lastly, we propose two different types of prior knowledge for the vision and language prompts to make it spatial sensitive and further enhance the model's generalisability.

# 3 Method

## 3.1 Overview

HOI Detection aims to detect all interactive human-object pairs and predict the interactive relationship for them. Formally, we define the interaction as a quadruple $(b_h, b_o, a, o)$: $b_h, b_o$ represent the bounding box of humans and objects and $a \in \mathbb{A}, o \in \mathbb{O}$ represent the human action and object category, where $\mathbb{A} = \{1, 2, ..., A\}$ and $\mathbb{O} = \{1, 2, ..., O\}$ denote the human action and object set, respectively. Then given an image $\mathbf{I}$, our goal is to predict all quadruples that exist in $\mathbf{I}$. To avoid struggling with multi-task learning (Zhang et al., 2021a) and missing potentially interactive human-object pairs, we divide the HOI detection task into two stages: human-object detection and interaction classification.

The overall architecture of our PID is illustrated in Figure 2. In the first stage, we use an off-the-shelf object detector D, e.g., DETR (Carion et al., 2020), and apply appropriate filtering strategies to extract all instances and exhaustively enumerate the detected instances to compose human-object pairs. Then in the second stage, we first encode the image $\mathbf{I}$ using a pretrained image encoder $\mathrm{E_I}$, i.e., $f_I = \mathrm{E_I}(\mathbf{I}) \in \mathbb{R}^{H \times W \times C}$. We define the union region $b_u$ as the smallest rectangular region that contains $b_h, b_o$. Then following the multi-branch architecture of previous HOI detection works (Hou et al., 2020; 2021b), we utilize $b_h$, $b_o$, and $b_u$ to extract features for the human branch, the object branch, and the interaction branch from the feature map $f_I$ via ROI-align (He et al., 2017), respectively.

To propagate knowledge from seen HOI categories to unseen HOI catetories and eliminate the dependence between the spatial-aware feature extraction and interaction classification tasks, as shown in Figure 2, we propose decoupled vision prompts $P_V$ and language prompts $P_L$ for the image encoder $\mathrm{E_I}$ and text encoder $\mathrm{E_T}$, respectively. We incorporate $P_V$ into the image encoder to adjust its capabilities from individual instance understanding to pair-wise relation understanding. We feed forward $P_L$ to get the weights of interaction classifier $W_L$, which we can then use to calculate the interaction score for the given human-object pair.

Furthermore, we introduce constraints for multi-modal prompts to alleviate the overfitting problem in prompt learning process, as shown in Figure 2. Specifically, we propose a lightweight neural network to generate input-conditioned vision prompts $P_V$, where instance-level visual prior $C_V$ is used as constraint. The conditional vision prompts are verb-agnostic and can alert the image encoder $\mathrm{E_I}$ to all the potential interactive instances in the image. For the linguistic side, to better incorporate language priors from the pretrained vision-language model, we use human-designed prompts $C_L$ to keep $P_L$ from diverging too much or overfitting to the seen categories through a regularization loss.

## 3.2 Multi-Modal Prompts

To efficiently fine-tune the multi-modal encoders for HOI detection task, we introduce multi-modal prompts to learn shared knowledge of seen and unseen classes while the entire pretrained parameters are kept frozen, as shown in Figure 2. We utilize vision prompts $P_V$ for extracting spatial-aware visual features and language prompts $P_L$ for interaction classification. To prevent error-propagation between visual representation and semantic interaction classification for the zero-shot HOI Detection task, we separate the visual and language prompts while maintaining alignment between the modalities' representations and allowing knowledge sharing between seen and unseen HOI classes.

**Language Prompts:** Language is the key to generalise for unseen HOI categories, especially for the unseen verbs setting, thanks to its natural generalisability. For the verb classes, we first convert them to the text descriptions through human-designed prompts. For example, given a verb of class $a$, we format it as "A photo of a person [Verb-ing] an object." We then tokenize the sentence for all verb classes $a \in \mathbb{A}$ to obtain $C_L^a$. We denote $U_L = [U_L^1, U_L^2, ..., U_L^S]$ as the learnable context words, where $S$ denotes the number of learnable prompts. The context words $U_L$ are shared among all classes and thus serve as a bridge between semantics of seen and unseen categories. The final representation of class $a$ can be obtained by concatenation of learnable context words $U_L$ and the text description's representation $C_L^a$:

$$P_L^a = \text{concat}(U_L, C_L^a), \tag{1}$$

where $P_L^a$ is the representation of class $a$ with learnable context words. Then the prototype of the class $a$ can be obtained by the text encoder $\text{E}_\text{T}$:

$$W_L^a = \text{E}_\text{T}(P_L^a), a \in \mathbb{A} \tag{2}$$

The prototypes should be the representative features belonging to the corresponding category. Given a sample, the similarity with a prototype could represent how likely it belongs to the category. After performing $l_2$-normalization on all prototypes $W_L^a$, the interaction classifier $W_L$ is then constructed from prototypes of all target classes' embeddings:

$$W_L = \text{concat}(W_L^1, W_L^2, ..., W_L^A) \tag{3}$$

**Vision Prompts:** Simply introducing the learnable language prompts is not enough for the HOI detection task since the large vision-language model is originally trained by a image-text matching problem. As a result, we introduce vision prompts for the image encoder to make it be aware of the spatial-aware relation recognition task and further extend its ability to determine whether the human-object pair in a given region is interactive. We use decoupled vision prompts to separate the spatial-aware visual feature extraction task from the interaction classification task and break error-propagation in the process. The vision prompts help adjusting visual representations to be interactiveness-aware and thus benefit recognizing unseen HOIs.

Specifically, we introduce vision prompts $P_V = [P_V^1, P_V^2, ..., P_V^M]$ that are independent of language prompts, where $M$ denotes the number of learnable vision prompts, which can grasp valuable verb-agnostic knowledge for generalising to unseen human-object triplets. The vision prompts $P_V$ are composed of $M$ learnable vectors with the same dimension. Because determining whether a human-object pair in a given union region is interactive has many similarities between seen and unseen HOI classes, we propose vision prompts to store such knowledge for zero-shot HOI detection. Furthermore, the vision prompts can implicitly learn spatial information to assist the pretrained image encoder $\text{E}_\text{I}$ in transitioning from instance-level recognition to pairwise relation detection task. However, fusing $P_V$ into the pretrained image encoder $\text{E}_\text{I}$ is not trivial since $\text{E}_\text{I}$ is simply pretrained by image-text matching task with an entire image as input. To deal with it, we propose a lightweight adapter LA that can help to fuse knowledge learned by prompts to $\text{E}_\text{I}$ via the cross-attention mechanism. LA can incorporate verb-agnostic prior knowledge into low-level feature maps of $\text{E}_\text{I}$ in order to capture the local spatial structures required for pair-wise relationship detection. Note that the vision prompts are interaction-agnostic and thus are naturally suitable for zero-shot generalisation. We denote $X_i \in \mathbb{R}^{hw \times d}$ as the feature map of i-th block of $\text{E}_\text{I}$. To keep the model efficient and avoid redundant information injection, we first down-project the feature dimension of $X_i$ to $d'$ ($d' \ll d$) through a simple MLP:

$$X_i' = \text{MLP}(X_i), \tag{4}$$

where $X_i'$ shares the same feature dimension with $P_V$. Then we inject context knowledge $P_V$ into $X_i$ through LA:

$$X_i = X_i + \text{MLP}(\text{LA}(X_i', P_V, P_V)), \qquad (5)$$

where LA is implemented with an attention mechanism and $X_i'$ is treated as query and $P_V$ is treated as key and value, as shown in Figure 3.

### 3.3 Conditional Multi-Modal Prompts

To alleviate the problem of overfitting to seen concepts in prompt learning process and further improve the model's generalisability, we propose two different types of prior knowledge for vision and language prompts, respectively. The independent vision and language prior knowledge can further help to eliminate the dependence between the two modalities. We use different methods to force constraints for vision and language prompts. For the vision prompts, we use input-conditioned prior knowledge as the direct constraint, whereas for the language prompts, we apply the constraint at the feature space through a regularization loss to better exploit the text space of the pretrained text encoder.

**Conditional Language Prompts:** To further utilize the feature space learnt by the text encoder of VLMs and improve generalisation for unseen classess, we propose to use human-designed prompts to constrain the features space of the learnable language prompts. The constraint ensures that prototypes of seen and unseen classes leave a reasonable separation margin among each other and do not diverge too far apart. We apply a regularization loss to reduce the discrepancy between the feature representation of $P_L$ and that of the human-designed language prompts $C_L$. Specifically, we encourage the soft prompt $P_L^i$ to be encoded close to its corresponding human-designed prompt $C_L^i$ through a contrastive loss. The Conditional Language Prompts Loss ($\mathcal{L}_{clp}$) can be formulated as:

$$\mathcal{L}_{clp} = -\sum_{i=1}^{A} \log \frac{\exp(\cos(W_L^i, W_{hum}^i))}{\sum_{j=1}^{A} \exp(\cos(W_L^i, W_{hum}^j))}, \quad (6)$$

where $W_{hum} = \text{E}_\text{T}(C_L)$ is the encoded features of human-designed prompts $C_L$ and $W_L$ is the feature representation of $P_L$.

**Conditional Vision Prompts:** To encourage the model to treat the seen and potentially unseen interactive instances equally, we utilize instance-level information as input-conditioned prior knowledge for vision prompts. We propose a projection network $\text{Proj}$ to explicitly con-

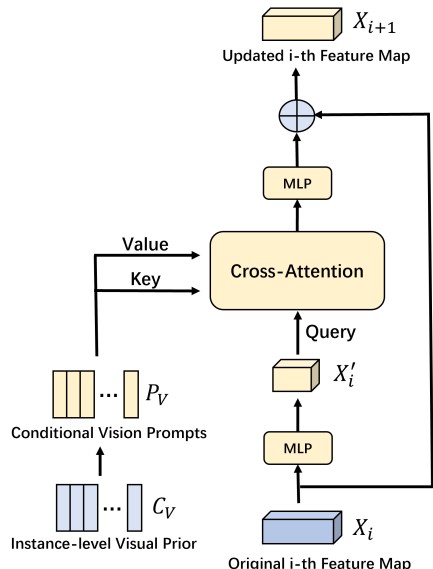

Figure 3: **Injection of conditional vision prompts.** To obtain spatial-aware visual features with conditional vision prompts, we inject the conditional vision prompts $P_V$ into the original feature map based on the attention mechanism.

dition $P_V$ on the human-designed prior knowledge, including (1) bounding boxes $b$, which captures the spatial information of detected objects. The spatial configuration might provide cues for understanding interactiveness and is good at transferring as it is instance-agnostic. (2) confidence scores $s$, which reflect the quality and uncertainty of the candidate instances. (3) semantic embeddings $e$ of the detected instances, which is obtained by CLIP language encoder and enables $P_V$ to leverage the category priors to capture which objects can be interacted with. These three types of prior knowledge reflect the detected instances' spatial configurations, quality, and semantic information, respectively. By incorporating these prior into the vision encoder, we can effectively guide it to extract features that are more attuned to interactiveness. We obtain the three components used in instance-level visual prior from the results of instance detection, which avoids groundtruth leakage and enables us to access and utilize these information during inference. Note that the semantic embeddings used here are first projected to the feature space of vision prompts and thus can be disentangled from the language prompts. Then the conditioned vision prompts $P_V$ can be formulated as the following:

$$P_V = \text{Proj}(\text{concat}(b, s, e)) \qquad (7)$$

Table 1: **Performance comparison for zero-shot HOI detection.** RF indicates rare first, NF indicates non-rare first. UC and UV denote unseen composition and unseen verb settings, respectively. PD denotes performance degradation. Our model not only outperforms previous methods on unseen classes by a wide margin, but it also exhibits the least performance degradation and thus exhibits greater generalisability across all three zero-shot settings.

| Method | Type | Unseen↑ | Seen↑ | Full↑ | PD↓ |
|---|---|---|---|---|---|
| ConsNet (Liu et al., 2020b) | UC | 16.99 | 20.51 | 19.81 | 3.52 |
| HOICLIP (Ning et al., 2023) | UC | 23.15 | 31.65 | 29.93 | 8.50 |
| *PID (Ours)* | UC | **29.60** | **32.39** | **31.84** | **2.79** |
| VCL Hou et al. (2020) | RF-UC | 10.06 | 24.28 | 21.43 | 14.22 |
| ATL Hou et al. (2021a) | RF-UC | 9.18 | 24.67 | 21.57 | 15.49 |
| FCL Hou et al. (2021b) | RF-UC | 13.16 | 24.23 | 22.01 | 11.07 |
| GEN-VLKT Liao et al. (2022) | RF-UC | 21.36 | 32.91 | 30.56 | 11.55 |
| EoID Wu et al. (2022a) | RF-UC | 22.04 | 31.39 | 29.52 | 9.35 |
| HOICLIP (Ning et al., 2023) | RF-UC | 25.53 | **34.85** | **32.99** | 9.32 |
| *PID (Ours)* | RF-UC | **28.82** | 33.35 | 32.45 | **4.53** |
| VCL Hou et al. (2020) | NF-UC | 16.22 | 18.52 | 18.06 | 2.30 |
| ATL Hou et al. (2021a) | NF-UC | 18.25 | 18.78 | 18.67 | 0.53 |
| FCL Hou et al. (2021b) | NF-UC | 18.66 | 19.55 | 19.37 | 0.89 |
| GEN-VLKT Liao et al. (2022) | NF-UC | 25.05 | 23.38 | 23.71 | **-1.67** |
| EoID Wu et al. (2022a) | NF-UC | 26.77 | 26.66 | 26.69 | -0.11 |
| HOICLIP (Ning et al., 2023) | NF-UC | 26.39 | 28.10 | 27.75 | 1.71 |
| *PID (Ours)* | NF-UC | **29.82** | **28.80** | **29.00** | -1.02 |
| GEN-VLKT Liao et al. (2022) | UV | 20.96 | 30.23 | 28.74 | 9.27 |
| EoID Wu et al. (2022a) | UV | 22.71 | 30.73 | 29.61 | 8.02 |
| HOICLIP (Ning et al., 2023) | UV | 24.30 | 32.19 | 31.09 | 7.89 |
| *PID (Ours)* | UV | **26.27** | **32.60** | **31.71** | 6.33 |

The $\texttt{Proj}$ network is implemented by a simple three-layer perceptron. The conditioned vision prompts $P_V$ explicitly provides more valuable prior knowledge for the image encoder to better transfer to unseen classes.

## 3.4 TRAINING PID

Based on the spatial-aware feature map $f_I$ and the extracted bounding boxes $b_h$, $b_o$, and $b_u$, we first apply ROI-Pooling to extract features for different branches:

$$f_{hum}, f_{obj}, f_{inter} = \text{ROI}(f_I, b_h), \text{ROI}(f_I, b_o), \text{ROI}(f_I, b_u) \tag{8}$$

The interaction classifier $W_L$ is composed of prototypes of all target classes as described in Section 3.2. We then calculate the action prediction $s_{ho}$ for the corresponding human-object pair as following:

$$s_{ho} = (\lambda_{hum}f_{hum} + \lambda_{obj}f_{obj} + \lambda_{inter}f_{inter})W_L^T \tag{9}$$

We incorporate the object confidence scores into the final scores of each human-object pair. We denote $\sigma$ as the sigmoid function. The final score $s_{ho}^{final}$ is computed as:

$$s_{ho}^{final} = \sigma(s_{ho}) \cdot (s_h)^\lambda \cdot (s_o)^\lambda, \tag{10}$$

where $s_h$ and $s_o$ are confidence scores given by object detector D, and $\lambda > 1$ is a constant that is used to suppress overconfident objects during inference. The whole model is trained on focal loss Lin et al. (2017) $\mathcal{L}_{cls}$ for action classification and language regularization loss $\mathcal{L}_{clp}$ at the same time. We use $\lambda_{clp}$ as the hyper-parameter weight. The whole loss is formulated as:

$$\mathcal{L} = \mathcal{L}_{cls} + \lambda_{clp}\mathcal{L}_{clp} \tag{11}$$

## 4 EXPERIMENTS

### 4.1 EXPERIMENT SETTING

**Dataset:** HICO-DET (Chao et al., 2018) is a dataset for detecting human-object interactions in images and has 47,776 images (38,118 in train set and 9,658 in test set) and is annotated with

<*human, verb, object*> triplets. 600 HOI categories in HICO-DET are composed of 80 object classes and 117 verb classes, including no interaction labels.

**Zero-shot Setups:** To validate our model's zero-shot performance, we evaluate our model on four zero-shot settings on HICO-DET: 1) Unseen Composition (UC), where the training data contains all categories of object and verb but misses some HOI triplet categories. 2) Rare First Unseen Combination (RF-UC) (Hou et al., 2021b), which prioritizes rare HOI categories when selecting held-out HOI categories. 3) Non-rare First Unseen Combination (NF-UC) (Hou et al., 2021b), which prioritizes non-rare HOI categories instead. Therefore, the training set of the NF-UC setting contains much fewer samples and thus is more challenging. 4) Unseen Verb (UV) (Liao et al., 2022), which is set to discover novel categories of actions and reflects a unique characteristic of zero-shot HOI detection.

**Evaluation Metric:** Following the common evaluation protocol, we use the mean average precision (mAP) to examine the model performance. A detected human-object pair is considered as a true positive if 1) both the predicted human and object boxes have the Interaction-over-Union (IOU) ratio greater than 0.5 with regards to the ground-truth boxes. 2) the predicted HOI categories are accurate.

## 4.2 IMPLEMENTATION DETAILS

We follow the standard protocol of existing zero-shot two-stage HOI detectors (Bansal et al., 2020; Hou et al., 2020) to fine-tune DETR on all the instance-level annotations of the entire HICO-DET dataset prior to training PID. We leverage ViT-B/16 backbone of CLIP in all experiments. The weight $\lambda_{clp}$ for the regularization loss is set to 2.0 during training. We use AdamW (Loshchilov & Hutter, 2017) as the optimizer with an initial learning rate of 1e-3 and train PID for only 15 epochs. Training is conducted with a batch size of 32 on 2 NVIDIA A100 devices.

## 4.3 EFFECTIVENESS FOR ZERO-SHOT HOI DETECTION

We evaluate the performance of our model and compare it with existing zero-shot HOI detectors under UC, RF-UC, NF-UC, and UV settings of HICO-DET (Chao et al., 2018) dataset.

As shown in Table 1, our model has demonstrated exceptional performance by outperforming all previous detectors by a significant margin on the unseen classes. Furthermore, our model performs comparably to the previous detectors on the seen classes, resulting in an overall outstanding performance. To be specific, compared to the previous state-of-the-art methods, our model achieves

Table 2: **Ablation on network modules on the Unseen Verb setting.** $Cond$ is short for conditional.

| Setting | Unseen | Seen | Full |
|---|---|---|---|
| $Base$ | 15.12 | 14.85 | 14.89 |
| $Base+P_L$ | 19.83 | 25.99 | 25.13 |
| $Base+Cond\ P_L$ | 20.56 | 26.03 | 25.27 |
| $Base+P_V$ | 18.83 | 32.78 | 30.83 |
| $Base+Cond\ P_V$ | 19.40 | **32.93** | 31.03 |
| $Base+P_L+P_V$ | 23.55 | 32.71 | 31.43 |
| $Base+Cond\ P_V+Cond\ P_L$ | **26.27** | 32.60 | **31.71** |

a relative mAP gain of 27.86%, 12.89%, 19.04% and 8.11% on unseen classes on four zero-shot settings, respectively. The performance gap demonstrates our model's ability to excel in both spatial relation extraction for visual features and prototype learning for interaction classification. Notably, since unseen classes under the NF-UC setting are sometimes more common and semantically straightforward, both our model and previous models (Liao et al., 2022; Wu et al., 2022a) may perform better on the unseen split than on the seen split.

Furthermore, previous methods exhibit severe performance degradation between seen and unseen classes, indicating a lack of generalisability. Our model, on the other hand, could alleviate the problem to a large extent and has a high potential for generalisation to previously unseen HOIs, confirming the effectiveness of our multi-modal prompts with constraints.

## 4.4 ABLATION STUDY

**Network Modules:** Here we study the effectiveness of different modules of PID under the unseen verb setting of HICO-DET. Experiments are conducted on the unseen verb setting. We consider

Table 3: **Ablation on the constraint of the language prompts.** Introducing conditional language prompts with appropriate weights contributes to enhancing the model's generalization capability.

| $\lambda_{clp}$ | Unseen | Seen | Full |
|---|---|---|---|
| 0 | 24.49 | 32.48 | 31.36 |
| 2.0 | **26.27** | **32.60** | **31.71** |
| 5.0 | 24.45 | 32.24 | 31.15 |

Table 4: **Ablation on backbone.** Our approach demonstrates a notable improvement in performance when combined with superior pre-trained models.

| Backbone | Type | Unseen | Seen | Full |
|---|---|---|---|---|
| ViT-B/16 | RF-UC | 28.82 | 33.35 | 32.45 |
| ViT-L/14 | RF-UC | **35.15** | **37.26** | **36.84** |
| ViT-B/16 | NF-UC | 29.82 | 28.80 | 29.00 |
| ViT-L/14 | NF-UC | **34.27** | **34.90** | **34.78** |
| ViT-B/16 | UV | 26.27 | 32.60 | 31.71 |
| ViT-L/14 | UV | **31.98** | **37.17** | **36.44** |

the model with CLIP's weights for initialization and no learnable parameters to be the baseline. As shown in Table 2, our $Base$ model achieves an mAP of 15.12% on unseen classes. Additionally, we observe four behaviors related to the use of prompts in the HOID task: (1) we find that when adding only one unconditional prompt, visual prompts $P_V$ results in a more significant gain as shown in the fourth lines in Table 2, especially for the seen classes, due to the fact that the original vision encoder is designed for image-level recognition rather than regional spatial-aware relationship recognition, which is required for our HOID task. (2) while adding only one unconditional prompt on baseline does not significantly enhance the recognition of unseen verbs, adding multi-modal prompts on top of that leads to a substantial improvement on unseen classes. Specifically, as shown in the sixth line in Table 2, our approach with unconditional multi-modal prompts achieves 23.55% mAP on unseen classes, outperforming many previous methods. This indicates that incorporating multi-modal prompts helps to establish a better semantic space that aligns with the regional spatial-aware visual space. (3) adding conditions independently on top of each prompt can provide some benefits, but the performance gains of the model are limited due to the inherent constraints of other frozen modalities. However, when these conditions are combined together, our model is able to achieve a clear gain, demonstrating its ability to leverage distinct types of prior knowledge for the subtasks of interactiveness-sensitive feature extraction and generalisable interaction classification.

**Constraints for Language Prompts:** The role of the Conditional Language Prompts Loss (contrastive learning) is to serve as a regularization term, allowing the text prompt $P_L$ to learn contextual information through learnable context $U_L$ while preventing excessive deviation from the CLIP text feature space, avoiding a decrease in generalization performance. We conduct experiments using various weights for the regularization loss, as presented in the Table 3. We observe that: (1) When changing the weight $\lambda_{clp}$, the changes in model performance are mainly shown in the unseen categories. This indicates that the regularization loss primarily affects the model's generalization ability. (2) When $\lambda_{clp}$ is set to 0, the lack of constraint in the text prompt might cause the textual features to deviate from the CLIP feature space, and decrease the performance on unseen categories. (3) As $\lambda_{clp}$ is increased to 2.0, the performance on unseen categories improves, demonstrating an enhancement in model generalization. However, further increasing $\lambda_{clp}$ could potentially result in $U_L$ becoming useless, constraining the model's capacity and leading to a decrease in the final performance. Therefore, introducing conditional language prompts with appropriate weights contributes to enhancing the model's generalization capability.

**Extension to Advanced Backbones:** Our approach can benefit from advanced pre-trained vision-text models, such as ViT-L/14, and we have conducted experiments using larger backbone networks. As pre-training of VLMs is in a quick-evolving stage, it's a desirable property of our model to be easily adaptable to new developments. To further ensure that our model can recognize HOIs even with small objects, we leverage ViT-L/14-336px to extract high-resolution feature maps. The results in Table 4 demonstrate that our approach shows a significant performance boost when incorporated with better pre-trained models, highlighting the great extension ability of our model.

## 5 CONCLUSION

We propose PID, a Prompt-based zero-shot human-object Interaction Detector. Our model separates zero-shot HOI detection into two subtasks: extracting spatial-aware visual features and interaction classification. PID deals with the subtasks with decoupled multi-modal prompts to break error-propagation in-between. Furthermore, PID employs appropriate constraints for each modality to reduce overfitting to seen HOI classes. Experiments on three zero-shot settings show that PID outperforms all pervious methods by a large margin and shows the least performance degradation, establishing a new state-of-the-art for zero-shot HOI detection.

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

## A  APPENDIX

This supplementary material provides additional details as well as more ablation study. We begin by describing the experiment setting for zero-shot HOI detection in Section A.1. Then we describe the input pipeline and hyper-parameters involved in the model in Section A.2. Then we provide further analysis on conditional vision prompt in Section A.3. At last, we provide qualitative results of our model on detecting novel HOIs in Section A.4.

### A.1  EXPERIMENT SETTING

Following Bansal et al. (2020), in the Unseen Combination (UC) setting, all action categories and object categories are included during training, but 120 classes of HOI triplets (i.e. combinations) are missing. Similiar to  Hou et al. (2020), in the Rare First Unseen Composition (RF-UC) and Non-rare First Unseen Composition (NF-UC) settings, 120 HOI classes are missing during training. The RF-UC selects unseen categories from tail HOIs preferentially, while the NF-UC prefers the head categories. The selected HOI categories are then removed from the training set. For the UV setting, 20 verbs are selected from all total 117 verbs to form 84 unseen and 516 seen HOIs following Liao et al. (2022).

### A.2  IMPLEMENTATION DETAIL

**Input Pipeline:** Input images are first randomly flipped horizontally and scaled such that the shortest side is at least 480 and at most 800 pixels. We then randomly jitter the color of the image on brightness, contrast saturation and hue with probability of (0.4, 0.4, 0.4), respectively. The images are fed into the detector at this point. We then resize the images to (224, 224) and feed them into the image encoder $E_I$.

**Object Detection:** In the first stage of our method, we first use an off-the-shelf object detector and apply appropriate filtering strategies to extract all instances. Specifically, We initialize the detector's weights from the publicly available model pretrained on MS COCO (Lin et al., 2014) and fine-tune it on the detection annotations of HICO-DET (Chao et al., 2018), following Zhang et al. (2022a); Wang et al. (2022a); Qu et al. (2022). We filter out detections with scores less than 0.2 and perform non-maximum suppression with a threshold of 0.5 to remove low-quality and redundant detections. Then, we reserve at least 3 and at most 15 boxes for humans and objects each for every image.

**Hyper-parameters:** For the language prompts $P_L$, we set the length of context words $S$ to be 16. For the vision prompts $P_V$ introduced in Section 3.2, we set the number of learnable vision prompts $M$ to be 10. We down-project the feature dimension to $d'(=64)$ when injecting prior knowledge into $E_I$. $\lambda$ is set to 1 during training and 2.8 during inference to suppress overconfident objects (Zhang et al., 2021b).

Table 5: **Ablation on different blocks to select.** PD denotes performance degradation.

| Selected blocks | Unseen | Seen | Full | PD |
|---|---|---|---|---|
| Low-level | 22.06 | 28.75 | 27.82 | 6.69 |
| High-level | 25.02 | 32.55 | 31.49 | 7.53 |
| All | **26.27** | **32.60** | **31.71** | **6.33** |

Table 6: **Ablation on different types of visual condition on the Unseen Verb setting.** $b, s, e$ represent bounding boxes, confidence scores and semantic embeddings for the detected instances, respectively.

| $b$ | $s$ | $e$ | Unseen | Seen | Full |
|---|---|---|---|---|---|
| - | - | - | 20.56 | 26.03 | 25.27 |
| ✓ | - | - | 24.70 | 32.39 | 31.31 |
| ✓ | ✓ | - | 24.81 | **33.20** | **32.02** |
| ✓ | ✓ | ✓ | **26.27** | 32.60 | 31.71 |

### A.3 FURTHER ANALYSIS ON CONDITIONAL VISION PROMPT

In this section, we exploit the importance of different blocks when fusing conditional vision prompt $P_V$ into the image encoder $E_I$: *All* means all blocks are selected; *Low-level* means to select the former blocks; *High-level* means to select the later blocks. As shown in Table 5, it's worth noting that choosing low-level blocks results in the least amount of performance degradation, while choosing high-level blocks results in the most. It demonstrates that low-level feature maps represent the model's transferability while high-level feature maps represent the model's discriminability. Selecting half of the blocks at random or all of the blocks results in medium performance degradation compared to selecting low-level blocks and selecting high-level blocks. Selecting all blocks brings the best performance on the unseen classes, demonstrating a good balance between transferability and discriminability.

We also study the effectiveness of different types of prior knowledge used as the constraints for $P_V$. As shown in Table 6, by simply using bounding boxes as condition brings a 4.14% mAP improvement on unseen classes, compared to the $Base$ model. Considering a scenario that involves a bicycle and a person beside it, actions such as "repair" or "inspect" are more likely to appear than "straddle" or "ride" due to the spatial configuration. Additionally, by adding confidence scores to the vision prior knowledge, our model performs better on unseen classes and achieves the best performance on the seen classes. This suggests that the confidence score serves as an indicator to help the model perform quality control by providing a measure of certainty for detected instances. Finally, the model achieves 26.27% mAP on unseen classes by further involving semantics into the vision prior, which proves improved interactiveness-aware feature extraction and better generalisability of our model. We treat all object semantics equally, regardless of whether they belong to seen or unseen HOI categories, which helps alleviate overfitting to seen HOI classes.

### A.4 QUALITATIVE RESULTS

As shown in Figure 4, we present several qualitative results of successful HOI detections. The visualized HOIs contain unseen verbs, e.g., the verb "wear" and "swing" which don't appear in the training set in the unseen verb setting. Our model successfully detects a human-wearing-tie triplet and a human-swing-baseball-bat triplet as shown in Figure 4a and Figure 4c, which shows powerful generalisability of our detector.

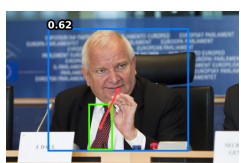 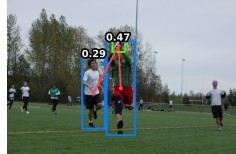 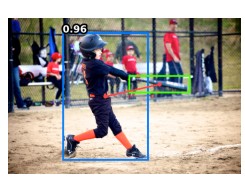 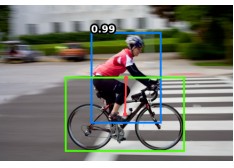

|     (a) wearing a tie | (b) blocking a frisbee | (c) swing a baseball bat | (d) ride a bike |

Figure 4: Visualization of successfully detected HOIs in the unseen verb setting. Each detected human-object pair is connected by a red line, with the corresponding interaction score overlaid above the human box. All the images contain unseen HOIs made up of unseen verbs and seen objects.

