# OpenReview forum: "Zero-shot Human-Object Interaction Detection via Conditional Multi-Modal Prompts"
_ICLR.cc/2024/Conference — Submitted to ICLR 2024_

### Official Review · Reviewer_n8e2 · 2023-10-23

**Soundness:** 3 good
**Presentation:** 3 good
**Contribution:** 2 fair
**Rating:** 5
**Confidence:** 5

**Summary:**

This paper proposed a prompt-based zero-shot HOI detector. It splits the detection task into two subtasks: extracting spatial-aware visual features and interaction classification. The vision and text prompts are jointly applied to the detector. Experimental results on the zero-shot settings show its effectiveness.

**Strengths:**

1. This paper is well written and organized. The vision and text prompts are also clearly explained.
2. Experimental results on the zero-shot settings demonstrate the effectiveness of the proposed method.

**Weaknesses:**

1. Overall, this work is very similar with the following ICCV2023 paper, including the overall framework, the conditional vision prompts and the learnable modules. What's the difference between the proposed method and the ICCV2023 paper.
A1: Efficient Adaptive Human-Object Interaction Detection with Concept-guided Memory, ICCV2023.
2. For the Lcls in (11), it is not clear how to connect the model with the GT labels.
3. This work only presents the HOI results using zero-shot settings. What's the result using the typical experimental settings?
4. Some important works from CVPR2023 are missing. Besides, the formats of some references are not consistent.

**Questions:**

Please see the Weaknesses.

---

> ### Author Response · Authors · 2023-11-20
>
> Dear Reviewer n8e2,
>
> First and foremost, we extend our deepest gratitude for your insightful feedback and hope our clarifications address your concerns. We're eager to highlight the significance and potential of our work.
>
> **1. The differences between the proposed method and the ICCV2023 paper.**
>
> Thank you once again for your careful consideration of our work.
> We propose an approach for zero-shot HOI detection that addresses the issue of **performance degradation between seen and unseen classes.** While the concurrent ICCV2023 paper focuses on improving efficiency when adapting CLIP to HOI detection, our method prioritizes generalizability. Our approach differs from it in several key aspects:
>
> + **Conditional Vision Prompts for Region-Level Interactiveness Knowledge:**
> While both the ICCV paper and our approach utilize prior knowledge from DETR, our method specifically focuses on leveraging instance-level visual priors to distinguish between non-interactive human-object pairs and interactive ones. This differentiation is crucial, especially in zero-shot HOID settings where unseen HOI concepts are encountered.
>
> + **Learnable Embeddings as Context Words:**
> To augment the CLIP text encoder's representation capabilities for actions, we introduce learnable embeddings as context words, i.e., an additional input known as language prompts.
>
> + **Regularization Loss for Generalizability:**
> Since the hand-engineered prompts outperform the learnable soft prompts for the zero-shot setting, then, in order to avoid overfitting to the seen classes and strengthen generalizability, we further propose that the learnable ones should be trained so that they can be correctly classified in language space where the class weights are given by the hand-engineered prompts. This is achieved through the **constrastive loss** in Equation (6) of the main text.
>
> + **Exclusive Reliance on Language for Interaction Classification:** For interaction recognition, the ICCV paper proposes a key-value mechanism storing both visual and linguistic knowledge. In contrast, we argue that visual knowledge is challenging to generalize to unseen classes. In other words, the visual features of unseen classes are hard to acquire. As a result, we exclusively employ language for interaction classification.
>
>
>
> **2. Connecting the predictions with the GT labels.**
>
>
> To associate the detected human–object pairs with the ground truth, we calculate the intersection-over-union (IoU) between each detected pair and the ground-truth pair. The IoU is computed for human and object boxes separately, and the minimum of the two is taken. Detected pairs are considered to be positive when the IoU surpasses a predefined threshold.
>
>
>
> **3. Experimental results using the typical settings.**
>
>
> We are truly grateful for this insightful consideration. To further validate the effectiveness of our method, we conduct experiments on the typical experimental settings on both the HICO-DET and V-COCO datasets. Empirically, we find that our method achieves competitive results on the default setting on both datasets. Notably, our method achieves sota on rare classes on HICO-DET, aligning with our methodology's design since rare classes are typically challenging to detect, similar to the scenario of detecting unseen interactions.
>
>
> |  Method  | Full  | Rare  | Non-Rare  |
> |   ----   | ----  | ----  |  ----   |
> | GEN-VLKT | 33.75 | 29.25 |  35.10  |
> | HOICLIP  | 34.54 | 30.71 |  35.70  |
> | Ours     | 34.26 | 32.22 |  34.86  |
>
>
> |  Method  | AP(Scenario 1) | AP(Scenario 2) |
> |   ----   | ----  | ----  |
> | GEN-VLKT | 62.4 | 64.4 |
> | HOICLIP  | 63.5 | 64.8 |
> | Ours     | 57.8 | 63.2 |
>
>
>
> **4. Missing related works.**
>
> Thank you for bringing this to our attention. We will incorporate discussions about the literature on HOI in CVPR2023 [1-4] into the related work section, and revise the formats of references in the final version.
>
> [1] Open-Category Human-Object Interaction Pre-Training via Language Modeling Framework
>
> [2] Relational Context Learning for Human-Object Interaction Detection
>
> [3] Category Query Learning for Human-Object Interaction Classification
>
> [4] ViPLO: Vision Transformer Based Pose-Conditioned Self-Loop Graph for Human-Object Interaction Detection

---

> ### Author Response · Authors · 2023-11-22
> **Further Discussion with Reviewer n8e2**
>
> Dear Reviewer n8e2,
>
> We sincerely appreciate the time you invested in reviewing our submission and your invaluable feedback. We have diligently addressed your comments and provided corresponding responses and results. We believe that these revisions have addressed the concerns you raised. We would be grateful for the opportunity to further discuss whether your concerns have been adequately addressed. If there are any aspects of our work that remain unclear, please do not hesitate to inform us.
>
> Once again, thank you for your guidance and insights.
>
> Warm regards,

---

### Official Review · Reviewer_c79G · 2023-10-30

**Soundness:** 3 good
**Presentation:** 2 fair
**Contribution:** 2 fair
**Rating:** 6
**Confidence:** 5

**Summary:**

The manuscript mainly focuses on the generalization of HOI detection, particularly zero-shot HOI detection. They proposed a Prompt-based HOI detection framework to improve the alignment between the visual and language representations with multi-modal prompts. Specifically, the decouple the visual and language prompts to improve spatial-aware feature learning. Meanwhile, they present several strategies to alleviate the overfitting to seen concepts. Effective experiments demonstrate the proposed method achieves a significant improvement on unseen categories.

**Strengths:**

1. The proposed visual-language decomposition strategy seems reasonable and demonstrates its effectiveness.
2. The proposed method demonstrates a significant improvement in zero-shot HOI detection based on large pre-trained models.
3. Part of the ablation experiment is beneficial for further research on visual relationship understanding. e.g. the effect of backbone networks.

**Weaknesses:**

Overall, the paper mainly borrows the popular adapt large models and prompt strategy for down-stream tasks. Considering that there are massive similar approaches in other fields, the novelty is limited. However, the reviewer still thinks it is beneficial for the development of zero-shot HOI detection. To some extent, the core idea is similar to CoOp and the following work Co-CoOp, though this paper also incorporates the visual prompts and has made some HOI-specific designs.

**Questions:**

1. The proposed method achieves smaller gap between seen and unseen category. According to Tab.1, PD is larger in RF-UC setting. Could you explain it? Moreover, do you have any ablation studies to check which module is more important for reducing the PD.
2. The paper aims to achieve verb-agnostic prior knowledge. Could you explain why the verb-agnostic feature is helpful for interactiveness-aware features? By the way, the local spatial structure is actually verb-dependent, e.g., different action pattern demonstrates different relative human-object positions. Thus, capturing local spatial structure seems to contradict to verb-agnostic representations.


In Table 4, the improvement on Unseen category is clearly better than seen category on RF-UC setting when you use a larger backbone network. Do you have any explanations?

---

> ### Author Response · Authors · 2023-11-20
>
> Dear Reviewer c79G,
>
> First and foremost, we extend our deepest gratitude for your thorough review and insightful feedback. Your recognition of our method is truly appreciated. We acknowledge the concerns you've raised and will attempt to address them point by point:
>
>
> **1. The novelty of our method.**
>
> Thank you once again for your careful consideration of our work.
> CoOp[1] first proposes to use context tokens as language prompts in the image classification task. Co-CoOp [2] proposes to explicitly condition language prompts on image instances. Recently, other approaches for adapting V-L models through prompting have been proposed. For example, ProGrad [3] utilizes gradient matching, while TTTuning [4] employs test time tuning.
>
> Different from them, our innovations mainly lie in the following two aspects:
>
> + **Introducing Multi-Modal Prompts:** Previous methods have mainly focused on unimodal solutions, which means adding prompts either in the vision or language branch, particularly in the language branch when adapting Vision-Language pretrained models. We have discovered that unimodal prompts are not effective for the HOID task. This is because the HOID task requires not only the generalizability of the text branch through language prompts but also the inclusion of vision prompts to enhance the vision encoder's understanding of region-level spatial relationships. In light of this, we first propose multi-modal prompts in zero-shot human-object interaction detection to improve visual-language feature alignment and zero-shot knowledge transfer.
>
> + **Leveraging Prior Knowledge for Vision and Language Prompts:** Previous methods involving conditional prompts, like CoCoOp, generate an input-conditional token for each image and only apply it in the text branch. Our method differs from it in the following two aspects: (1) First, we propose a vision condition that incorporates spatial information for each image, making it more fine-grained and suitable for the HOID task in the vision branch. (2) Second, we introduce a language condition as a regularization term to prevent the model from deviating excessively from the CLIP text feature space.
>
>
> [1] Zhou, Kaiyang, et al. "Learning to prompt for vision-language models."
>
> [2] Zhou, Kaiyang, et al. "Conditional prompt learning for vision-language models."
>
> [3] Zhu, Beier, et al. "Prompt-aligned gradient for prompt tuning."
>
> [4] Shu, Manli, et al. "Test-time prompt tuning for zero-shot generalization in vision-language models."
>
>
>
> **2. Discussion on the PD.**
>
> + **PD is larger in RF-UC setting.**
>
> The PD could be attributed to the semantic gap between the seen and unseen classes. In RF-UC setting, the rare classes are filtered out during the training process. Given that rare classes in real-life scenarios are inherently infrequent and semantically challenging, they are not well-understood by the original CLIP text embedding space, leading to a larger PD.
>
>
> + **Ablation studies on which module is more important for reducing the PD.**
>
> We propose a constraint for each modality to alleviate the performance degradation. Following the reviewer's suggestion, we further conduct ablation studies to check which condition is more important for reducing the PD in RF-UC setting. As shown in the table below, we find that the language condition holds greater significance for reducing PD, showing the efficacy of utilizing hand-crafted prompts for regularization.
>
>
> |         Method         | Unseen | Seen | Full | PD |
> |          ----          | ----  | ----  |  ---- |  ---|
> |  w/o Vision Condition  | 27.97 | 32.95 | 31.95 | 4.98 |
> | w/o Language Condition | 26.27 | 32.60 | 31.71 | 6.33 |
> |      PID (Ours)        | 28.82 | 33.35 | 32.45| 4.53 |
>
>
> **3. Verb-agnostic knowledge for interactiveness-aware feature extraction.**
>
>
> + Verb-agnostic knowledge is independent of the specific actions or verbs involved in a given situation. Similarly, interactiveness judgment is inherently binary, devoid of the necessity for specific verb-related information. As a result, we propose to utilize verb-agnostic knowledge to facilitate interactiveness-aware feature extraction.
>
>
> + We would like to kindly remind you that the local spatial structure we utilize only involves the spatial information of the objects instead of the spatial arrangement of human-object pairs. Since an interaction may manifest at virtually any position within an image (taking into account the different shooting angles of the images), we argue that the local spatial structure is inherently verb-agnostic.

---

> > ### Comment · Reviewer_c79G · 2023-11-22
> >
> > ### PD
> > CLIP has a large language corpus. The rare first categories that are selected from the tail in HOI categories in HICO-DET might not be rare in CLIP. All those verbs and objects in HICO-DET are usually common in HICO-DET. It is an interesting phenomenon.

---

> ### Author Response · Authors · 2023-11-22
> **Further Discussion with Reviewer c79G**
>
> Dear Reviewer c79G,
>
> We sincerely appreciate the time you devoted to reviewing our manuscript and the invaluable feedback you provided. We have diligently addressed your comments and provided corresponding responses. We believe that these responses adequately address the concerns you raised. If any ambiguity remains, we sincerely invite further inquiries. We genuinely appreciate your time and dedication to reviewing our research.
>
> Once again, thank you for your constructive insights.
>
> Warm regards,

---

> ### Comment · Reviewer_c79G · 2023-11-22
>
> Dear authors,
> Thanks for your response and sorry for getting you late.
> ### novelty
> your response does actually support my point of view: the novelty is limited and the proposed method mainly adopts the core idea from CoOp by extending it to multi-modal prompts.
>
> > We have discovered that unimodal prompts are not effective for the HOID task.
> According to the value in the Table above, this is clearly not right. The method with language prompt only can actually achieve an effective result, which is 27.97 on RF-UC. right?
>
> ### verb-agnostic knowledge.
> How do you represent the local spatial structure? two binary maps? If so, I disagree that the spatial structure is verb-agnostic though I think it is helpful for interactioness recognition.

---

> ### Author Response · Authors · 2023-11-22
>
> Dear Reviewer c79G,
>
> Thank you for providing further feedback.
>
>
> **1. Unimodal prompts are not effective for the HOID task.**
>
> + Regarding the first line "w/o Vision Condition" in the table, we apologize for the confusion caused by the formulation. It actually means that we utilize both **unconditional vision prompts and conditional language prompts**, which constitutes a multi-modal setup. To align with the formulation in Table 2 of the main text, we have revised the table as follows:
>
> |         Method         | Unseen | Seen | Full | PD |
> |          ----          | ----  | ----  |  ---- |  ---|
> | $Base$ + $P_V$ + $Cond$ $P_L$ | 27.97 | 32.95 | 31.95 | 4.98 |
> | $Base$ + $Cond$ $P_V$ + $P_L$ | 26.27 | 32.60 | 31.71 | 6.33 |
> |  $Base$ + $Cond$ $P_V$ + $Cond$ $P_L$  | 28.82 | 33.35 | 32.45| 4.53 |
>
>
> + We would also like to kindly remind you that we discuss the performance when only utilizing unimodal prompts in Section 4.4 of the main paper. The ablation in Table 2 of the main text demonstrates that when using unimodal prompts, the model exhibits inferior performance on unseen classes.
>
>
> **2. Regarding the verb-agnostic knowledge.**
>
> To represent the local spatial structure, we solely utilize the bounding box coordinates of the objects.
> Based on your feedback, we will rename the verb-agnostic knowledge in the final version.
>
>
> Thank you once again for your valuable feedback.

---

> > ### Comment · Reviewer_c79G · 2023-11-23
> >
> > Thanks for your reply. You have marginally relieved my concerns about verb-agnostic knowledge. the bounding box coordinates might indicate different kinds of verb classification, e.g. ride horse and feed horse, which have actually been the basic features for verb discrimination. Therefore, verb-agnostic knowledge is unsuitable for this point. interactiveness-aware might be more accurate.

---

### Official Review · Reviewer_YqBQ · 2023-11-01

**Soundness:** 2 fair
**Presentation:** 2 fair
**Contribution:** 2 fair
**Rating:** 8
**Confidence:** 4

**Summary:**

In this submission, the authors tackled the problem of zero-shot human-object interaction (HOI) detection, which aims to localize and classify all the potential human-object interactions in a given image. The zero-shot setting for HOI detection further requires the model to detect novel classes of objects and/or actions which are not seen during training. Inspired by the recent trend on leveraging vision foundation models for HOI detection, the authors proposed a novel prompt learning based approach called PID. Specifically, several vision and language prompts are adopted to enhance the visual feature extraction and interaction classification, respectively. Some optimization tricks are also explored to prevent overfitting. Experimental results on HICO-DET partially show the significance of the proposed method.

**Strengths:**

1. Overall, the manuscript is well-written and easy to follow.
2. The use of prompt learning for HOI problems is a good direction to explore (and also a trend in computer vision).

**Weaknesses:**

1. The whole framework seems like a combination of multiple existing modules, e.g., DETR, CoOp/CoCoOp style prompts. The novelty and the motivation behind each of the design are unclear.
2. In 4.2, the authors mentioned that the DETR used for detecting all the humans and objects in the first stage is fine-tuned on the whole HICO-DET dataset. Does the 'whole' here mean both training and validation sets? If so, this is a weird setting as previous works (including Bansal et al. 2020 and Hou et al. 2020 that the authors claimed) never fine-tuned their detectors on the validation set, which would lead to extremely unfair comparison since the detector can significantly affect the overall performance.
3. The experiments are only conducted on a single dataset. Why the method is not tested on V-COCO?
4. The conclusion part lacks objective reflections on the deficiencies of this study and future prospects for improvements.

**Questions:**

See the weaknesses part. I'll consider changing the score after reading the authors' responses.

**Details Of Ethics Concerns:**

No ethics review is needed.

---

> ### Author Response · Authors · 2023-11-20
>
> Dear Reviewer YqBQ,
>
> First and foremost, we would like to express our sincere appreciation for your comprehensive review and valuable feedback. We are truly grateful for your recognition of our approach. Herein, we provide a detailed response to each of your concerns:
>
>
> **1. The novelty and the motivation behind each module.**
>
> We concur with your perspective that leveraging vision foundation models for the HOI detection task is a recent trend. In this work, we aim to apply CLIP for the HOI detection task through decoupled multi-model prompts:
>
> + **Conditional Vision Prompts for Region-Level Interactiveness Knowledge:** Since CLIP is originally designed for image-level recognition, we design vision prompts for extracting feature maps with region-level pair-wise interactiveness knowledge. Leveraging the instance-level prior knowledge from DETR, our encoder is guided to allocate heightened attention to regions with potential interactions.
>
> + **Learnable Embeddings as Context Words:** To augment the CLIP text encoder's representation capabilities for actions, we introduce learnable embeddings as context words, i.e., an additional input known as language prompts.
>
> + **Regularization Loss for Generalizability:** In order to utilize the original structure of CLIP's text embedding space and avoid overfitting to the seen classes, we further introduce a regularization loss. This loss prevents the model from becoming too specialized on the seen classes and strengthens its generalizability when dealing with unseen classes.
>
>
> **2. The data split used for fine-tuning DETR.**
>
> To clarify, in our approach, we fine-tune DETR **only on the training set**. The 'whole' in 4.2 means all the instance-level annotations, without distinction for instances engaged in unseen interactions, which is the same as previous works (including Bansal et al. 2020 and Hou et al. 2020). We apologize for the confusion caused by the earlier statement and will revise it in the final version of our work.
>
>
>
>
> **3. Experimental dataset.**
>
> We do not conduct experiments on v-coco dataset since the official evaluation utilities do not support a zero-shot setting.
>
> To further validate the effectiveness of our method, we conduct experiments on the typical experimental settings on both the HICO-DET and V-COCO datasets. Empirically, we find that our method achieves competitive results on the default setting on both datasets. Notably, our method achieves sota on rare classes on HICO-DET, aligning with our methodology's design since rare classes are typically challenging to detect, similar to the scenario of detecting unseen interactions.
>
>
> | Method   | Full  | Rare  | Non-Rare  |
> |   ----   | ----  | ----  |  ----   |
> | GEN-VLKT | 33.75 | 29.25 |  35.10  |
> | HOICLIP  | 34.54 | 30.71 |  35.70  |
> | Ours     | 34.26 | 32.22 |  34.86  |
>
>
> | Method | AP(Scenario 1) | AP(Scenario 2) |
> |   ----   | ----  | ----  |
> | GEN-VLKT | 62.4 | 64.4 |
> | HOICLIP  | 63.5 | 64.8 |
> | Ours     | 57.8 | 63.2 |
>
>
>
> **4. Reflections on the Study's Deficiencies and Future Directions.**
>
> Thank you for your constructive feedback. In light of your feedback, we will expand our conclusion section to incorporate the following points:
>
> + **Deficiencies:** In this paper, we employ CLIP for HOI detection via conditional prompt learning. However, a limitation of this approach is its dependency on the input resolution of CLIP, specifically constrained to 224 $\times$ 224 pixels, which may impede the accurate detection of interactions involving small objects.
>
> + **Future Prospects:** Considering the quick-evolving trends in Vision Language Models (VLM), and Large Language Models (LLM), in future, there are several promising directions: (1) We aim to employ VLMs for **flexible input resolutions** to deal with HOIs with different scales or distances. (2) We will explore to **integrate the semantic spaces of VLMs and LLMs** to improve the model's comprehension of interactions, especially for unseen interactions, aiming for a more generic HOI detection system in the real-world scenario. (3) We will explore efficient **knowledge transfer** methods in the future to facilitate the integration and collaboration of specialized HOI detectors powered by VLMs or LLMs, thereby reducing the inference costs of large models.

---

> > ### Comment · Reviewer_YqBQ · 2023-11-20
> >
> > Thanks for the reply from the authors. The response has addressed all my concerns. I would like to raise my rating to 8.

---

### Official Review · Reviewer_NGK6 · 2023-11-08

**Soundness:** 2 fair
**Presentation:** 2 fair
**Contribution:** 2 fair
**Rating:** 5
**Confidence:** 4

**Summary:**

This paper addresses the problem of zero-shot HOI detection with the key idea of using conditional multi-modal prompts. Specifically, the language prompts consist of two parts, human-designed prompts and learned ones, with the former being responsible for guiding the learning of the latter. The vision prompts is learned from instance-level visual priors, including bboxes, confidence scores, and semantic embeddings. The proposed method achieves competitive performance on HICO-DET.

**Strengths:**

The motivation is reasonable and the results are competitive.

**Weaknesses:**

**1. Lack of analysis.**
*1)* The language prompts are initialized as the concatenations of  $C_L^a$ and $U_L$, which are subsequently forced to be close to $C_L$, why? In this way, why not just using $C_L$ as language prompts? \
*2)* Does $\mathbb{A}$ contain unseen verbs? If that so, can this model recognize HOIs that are not present in HICO-DET? In other words, if I want to detect a HOI using this model, does the corresponding interaction verb have to be included in $\mathbb{A}$?  \
*3)* For vision prompts, where do the instance-level visual priors come from? Are they extracted by the pre-trained DETR? \
**2.** Actually, I do not understand why the vision prompts are useful for zero-shot HOI detection. Concretely, the visual feature extracted in this model do not seem to be very sepcial compared to the that in most two-stage based HOI detectors. \
**3.** It is unclear that how these prompts work?

**Questions:**

See weakness.

---

> ### Author Response · Authors · 2023-11-20
>
> Dear Reviewer NGK6,
>
> We greatly appreciate your thoughtful review and feedback. Herein, we provide a detailed response to each of your concerns:
>
>
> **1. More Analysis or Clarification.**
>
> **a) Regarding the language prompts:** We are truly grateful for this insightful consideration and conduct more ablation studies on the language prompts in the RF-UC setting as shown in the table below. We observe that:
>
> + Directly using $C_L$ as language prompts results in inferior performance on all categories since the original text embedding space is not optimized or tailored for these specific interactions.
>
> + By incorporating $U_L$, the overall performance improves about 1\% mAP at the cost of increased performance degradation between seen and unseen classes. This shows that while $U_L$ enhances the semantic space for interaction recognition, it also leads to a larger discrepancy between known and unknown classes.
>
> + In order to avoid the potential overfitting to seen classes and strengthen generalizability, $C_L$+$U_L$ should be trained so that they can be correctly classified in language space where the class weights are given by the hand-engineered prompts $C_L$.
>
>
> | Language Prompts | Unseen | Seen | Full | PD |
> |       ----       | ----  | ----  |  ---- |  ---|
> |      $C_L$       | 27.17 | 31.61 | 30.72 | 4.44 |
> |    $C_L$+$U_L$   | 26.27 | 32.60 | 31.71 | 6.33 |
> | $C_L$+$U_L$(Conditioned on $C_L$) | 28.82 | 33.35 | 32.45| 4.53 |
>
>
> **b) The capability of our method:** Following the previous experimental protocol, $\mathbb{A}$ contains both seen and unseen verbs. However, it is worth noting that since our method treats verb labels in a semantic manner, embedding them into a unified visual-and-text space as opposed to employing traditional one-hot labels, our method can detect any HOI in the wild given its name.
>
>
> **c) The instance-level visual priors:** Yes. The instance-level visual priors come from the pre-trained DETR.
>
>
>
> **2. Discussion of the vision prompts.**
>
> In contrast to most previous two-stage HOI detectors, our approach integrates prior knowledge from the pretrained detector directly into the image encoder's architecture, in addition to applying it to the output of the image encoder through ROI-Align. This integration **empowers the vision encoder with a heightened awareness of regions with greater potential for interactions**, thereby guiding the image encoder to produce feature maps that are more sensitive to interactiveness, which is crucial for zero-shot HOI detection.
>
>
>
> **3. Explanation on how these prompts work.**
>
>
> We address the challenge of HOI detection by dividing the task into two subtasks: visual feature extraction and interaction classification. We introduce vision prompts and language prompts to guide each subtask, leveraging prior knowledge as constraints to mitigate the issue of overfitting. The key ideas can be illustrated in the following 3 aspects:
>
> + **Conditional Vision Prompts for Region-Level Interactiveness Knowledge:** For visual feature extraction, we design conditional vision prompts that extract feature maps with region-level pair-wise interactiveness knowledge. Our encoder is guided to allocate heightened attention to regions with potential interactions, using instance-level prior knowledge from DETR.
>
> + **Learnable Embeddings as Context Words:** To augment the CLIP text encoder's representation capabilities for actions, we introduce learnable embeddings as context words, i.e., an additional input known as language prompts.
>
> + **Regularization Loss for Generalizability:** Since the hand-engineered prompts outperform the learnable soft prompts for the unseen classes, in order to utilize the original structure of CLIP's text embedding space, we further introduce a regularization loss. This loss prevents the model from becoming too specialized on the seen classes and strengthens its generalizability when dealing with unseen classes.

---

> ### Author Response · Authors · 2023-11-22
> **Further Discussion with Reviewer NGK6**
>
> Dear Reviewer NGK6,
>
> We sincerely appreciate the time and effort you have dedicated to reviewing our submission. We have carefully addressed your comments and provided corresponding responses and results. We believe that these responses and results adequately address your concerns. We would value an opportunity to discuss further whether your reservations have been resolved. Should there remain any aspects of our work that are unclear to you, please do not hesitate to inform us.
>
> Once again, thank you for your invaluable feedback.
>
> Best,

---

### Meta-Review · Area_Chair_oWEN · 2023-12-17

**Metareview:**

This paper addressed the zero-shot HOI problem via prompt learning. But the model idea borrows the popular adapt large models and prompt strategy for down-stream tasks, as many existing works did. Especially, Reviewer n8e2 mentioned the similarity with ICCV 2023 paper. And the paper did not explain why these prompts work. Thus the ACs decided to reject it.

**Justification For Why Not Higher Score:**

The novelty is limited.

**Justification For Why Not Lower Score:**

N/A

---

### Decision · Program_Chairs · 2024-01-16

Reject